# Food-Evoked Emotions and Optimal Portion Sizes of Meat and Vegetables for Men and Women across Five Familiar Dutch Meals: An Online Study

**DOI:** 10.3390/foods12061259

**Published:** 2023-03-16

**Authors:** Maria Isabel Salazar Cobo, Gerry Jager, Cees de Graaf, Elizabeth H. Zandstra

**Affiliations:** 1Division of Human Nutrition & Health, Wageningen University & Research, 6708 WE Wageningen, The Netherlands; 2Unilever Foods Innovation Centre Wageningen, 6708 WH Wageningen, The Netherlands

**Keywords:** portion size, food-evoked emotions, expected liking, expected satiety, meat, vegetables

## Abstract

Portion size manipulation is well known to be effective in increasing vegetable intake in adults, whereas less is known about the effects of portion size manipulation on reducing meat intake. This online study investigated the effects of recommended and regularly consumed portion sizes of vegetables and meat in five familiar Dutch meals. Participants evaluated 60 food pictures of five meals and used a 100 mm VAS to measure expected liking, satiety, food-evoked emotions, and the perceived normal portion size. The results show that both regular and recommended portions scored above 55 on the 100 mm VAS on expected liking and satiety. Similarly, both portion sizes scored high (55–70 on the 100 mm VAS) in positive emotions (i.e., happy, relaxed, and satisfied). Regarding the perceived amount of meat, men consistently preferred larger portions of meat than women. However, the optimal portion sizes of vegetables were similar for men and women. Furthermore, the recommended portion sizes led to positive food-evoked emotions, implying that the effective implementation of portion size strategies for increasing vegetable and limiting meat intake requires a careful, holistic approach focusing on the sensory characteristics of food products as well as the emotions evoked by the total food experience.

## 1. Introduction

In large parts of the industrialized world, people consume fewer vegetables and more meat than what is recommended [1,2]. The results of a ten-year cohort study in 18 countries in North and South America, Europe, the Middle East, Asia, and Africa show that adults do not meet the World Health Organization (WHO) recommendation of 400 g (i.e., five portions) of fruit and vegetables per day [3]. For example, in the Netherlands, the results of 2012–2016 Food Consumption Survey indicate that adults between 19 and 79 years consume about 131 g/day of vegetables [4], which is well below the recommended 200 g/day [5]. In terms of meat consumption, the literature reports a global increase of 58% in meat intake over 20 years as of 2018 in the United States, China, and Australia [6]. Moreover, it is expected that global meat consumption increases by 14% over the next decade compared with the base period average of 2018–2020 [7]. On the other hand, global concern over the consumption of animal products and awareness of meat intake reduction have led to an increased interest in plant-based diets or changes in eating patterns [8,9,10]. For example, in the United States, about 4% of adults reported being vegan or vegetarian between 2019 and 2020, whereas about 46% reported to sometimes or always eat vegetarian or vegan when eating out [11]. In addition, in the Netherlands, red meat consumption decreased from an average of 82 g per day to 79 g per day when comparing the results of the consumption surveys of 2012–2016 [4] and 2019–2020 [12]. Interestingly, the consumption of poultry increased from an average of 16 g per day to 18.4 g per day in the same period [4,12].

It is well known that an adequate consumption of fruit and vegetables lowers blood pressure and reduces the risk of cardiovascular diseases (CVDs) and type 2 diabetes. In contrast, the high consumption of red meat products is associated with higher risks of CVDs, diabetes, and colorectal and lung cancers [13,14]. The reduction in the prevalence of these diseases is high on national health agendas worldwide [13]. As a result, governments and health institutions promote fruit and vegetable intake and recommend reducing red and processed meat consumption. However, efforts from governments to implement these healthy eating strategies have not achieved the desired effects so far [15]. 

In order to be effective, public health interventions should not negatively affect the pleasure, satisfaction, and other positive emotions that people experience during food consumption. Assessing the emotional associations with food products is now becoming a key component in designing food products that satisfy consumer expectations [16,17]. In addition, emotional profiles make it possible to differentiate products with similar sensory profiles and hedonic ratings [17,18]. As a result, food-evoked emotions help to better understand consumer food choices. In recent years, a substantial body of work has shown that food choice may be, in part, guided by food-evoked emotions [19,20]. For example, Gutjar, et al. [21] showed that positive food-evoked emotions are strongly correlated with food choice (*r* = 0.88). 

One promising and effective strategy to change nutrition behavior is to modify food portion sizes [22]. The results of previous studies suggest that it is possible to reduce the portion size of foods and still report high satisfaction and liking [23,24]. Furthermore, the consensus of studies by Rolls and co-workers showed that increasing the portion size of low-energy-dense foods such as vegetables is an effective strategy to decrease overall energy intake [25,26,27]. In addition, a previous study by Roe et al. [26] on children aged between 3 and 5 years old found that substituting part of the portion of a meal with vegetables increased the vegetable intake by 41% and decreased the energy intake by 6% compared with the control group. Similar conclusions were drawn in the study by Carstairs et al. [25] on children aged between 3 and 5 years old, where the portion size of the high-energy-dense component of a meal was reduced by 60% while varying the accompanying vegetable items. These findings suggest that modifying the food portion size of different components in a meal is a potential effective strategy to increase vegetable intake. However, less is known about the effectiveness of portion size strategies for reducing meat consumption.

In two recent real-life restaurant setting studies, the portion sizes of vegetable and meat components were modified in a meal [28,29]. The first study showed that increasing vegetable portions in combination with decreasing meat portions (with customers being unknowing) increased the amount of vegetables consumed and decreased the amount of meat consumed [28]. Interestingly, despite the changes in portion sizes, participants remained satisfied with their restaurant visit and served meal. In follow-up research, Reinders et al. [29] again assessed consumption and satisfaction before and after changing the proportions of meat and vegetables in different meals in separate samples of restaurant guests in the Netherlands. As in the first study, the authors observed consistent results in satisfaction: a 12–34% decrease in the amount of meat and a 31–237% increase in vegetables did not alter guest satisfaction. Reinders et al. [29] stress the need to further investigate optimal amounts of meat and vegetables for different types of meals. It still remains to be investigated how changes in portion sizes of vegetables and meat influence satiety and food-evoked emotional responses to various meals with variable amounts of meat and vegetables. 

Regarding the portion sizes of meat and vegetables, the literature suggests that men seem to be more attached to meat [4,30,31], whereas women may have more positive attitudes towards vegetables [4,31,32]. This may imply that the effects of meat and vegetable portion manipulation on food-evoked emotions may differ by gender.

This online study investigated expected liking, satiety, and emotional responses to images of five different familiar Dutch meals with different amounts of vegetables and meat, i.e., regular amounts versus amounts adapted to meet the Dutch dietary guidelines [5]. Participants were presented with images of five different dinner meals (e.g., a meal consisting of green beans, boiled potatoes, and beef) varying in the portions of vegetables and meat (regular and recommended portions). The regular portion represented the amounts of meat and vegetables typically eaten with a warm meal in the Netherlands. Conversely, the recommended portions contained the amounts of meat and vegetables according to the Dutch dietary guidelines [5]. We hypothesized that the recommended portions of meat products (i.e., less meat than the regular portions) combined with increased amounts of vegetables would result in stronger negative emotions than the regular portions of meat because of the attachment of males to meat products. Similarly, we expected significant gender differences in the perceived and preferred amount of meat and vegetables. For expected satiety and liking, we expected no differences between regular and recommended portions of meat and vegetables.

Experiments have shown that modifying the portion sizes of meat products can change the food behaviors of consumers without affecting their satisfaction [29,33]. In addition, a recent study found that in the USA, consumers are interested in eating less red meat. The study explains that consumers prefer to eat smaller portion sizes of red meat instead of eating meat less often [34]. In terms of gender differences in meat intake, studies have found that men prefer larger portions. Hence, this online study aimed to respond to the following research questions:(a)Is there any preference for the regular or recommended portion size of the offered meals?(b)Is there any difference between regular and recommended portions in the intensity of positive and negative food-evoked emotions?(c)Is there any gender difference in the perception of the amount of meat and vegetables?

## 2. Materials and Methods

### 2.1. Participants 

Subjects were recruited by a market research agency, Essensor BV, in the Netherlands, with the aim to collect data from 270 participants, 50% male and 50% female, with 33% being aged between 18 and 35 years, 33% being aged between 36 and 50 years, and 33% being aged between 51 and 65 years. Subjects reported to be meat eaters or flexitarians and were from the areas of Utrecht, Wageningen, and Vlaardingen (the Netherlands), 33% from each region. Twelve participants were excluded from data analysis as they did not fully complete the questionnaire. The remaining group consisted of 258 participants (131 females and 127 males) with a mean age of 38.4 ± 13.6 years and mean BMI of 25.6 ± 4.5 kg/m^2^. Social Science Ethical approval was granted by the Social Science Ethics Committee of Wageningen University in February 2021. 

### 2.2. Food: Five Dinner Meals

Five dinner meals were used as food, namely, (a) green beans, boiled potatoes, and beef; (b) green peas with carrots, boiled potatoes, and chicken fillet; (c) broccoli, pasta, and Bolognese sauce with minced meat; (d) carrots, mashed potatoes, and white fish fillet; and (e) pizza with tomatoes, bell peppers, and salami. These dinner meals were chosen based on the familiarity of the target participants with the meals. Although one of the meals had white fish as an animal-based protein source, for the sake of readability, it will be referred to as the “meat” component of the meal, unless results differ in a meaningful manner between meat and fish.

### 2.3. Portion Sizes of Food

For both meat and vegetables, we used five portion sizes: extra-small, small, medium, large, and extra-large (Table 1). The portion sizes of meat and vegetables ranged from 25% less to 25% more than the recommended amounts of meat and vegetables. 

The *recommended* portion sizes of both meat and vegetables corresponded to the recommended intake of meat and vegetables as recommended by The Health Council of the Netherlands 2015 [5]. For meat ingredients, the recommended portion corresponded to the “small” portion in Table 1, i.e., 100 g of beef, chicken, minced meat, and white fish, and 25 g of salami. Meanwhile, for vegetables, the recommended portion corresponded to the “large” portion in Table 1; this is 200 g of vegetables.

The *regular* portion sizes of meat and vegetables were calculated on the basis of the average amount of vegetables consumed per day according to Dutch Food Consumption Survey 2012–2016 [4], Recommended Guidelines for Meat and Vegetables 2015 [5], and the average portions of meat (pre-packed) commercially available in local supermarkets. 

The portion size of carbohydrates, i.e., boiled potatoes, mashed potatoes, and pasta, corresponded to one regular serving for one adult, for instance, 150 g of boiled/mashed potatoes, 150 g of mashed potatoes, and 60 g of pasta, and these amounts were kept constant for all portions in each meal. For pizza, the portion size was the commercially available 320 g salami pizza (brand: Dr. Oetker Ristorante^®^). 

### 2.4. Food Pictures 

We used 60 pictures of foods, representing the different portion sizes of meat, vegetables, and the complete meals. Because this study was conducted during the first wave of the COVID-19 pandemic, the pictures were taken at home by following the protocol for food images as described by Charbonnier and colleagues [35]. We built a 60 × 60 × 60 cm photo studio, i.e., a cubic photo tent made from 2 cm thick Styrofoam to ensure the rigidity of the structure and internally covered with snow-white fabric to soften and reflect the light. A plain, gray color background (stone gray 300 g/m² photo cardboard; brand, Folia) was set to create a good contrast between the plate and the background. To take the food pictures, a high-resolution, digital, single-lens reflex Nikon camera was used. The camera was mounted on a tripod to ensure a height of 38 cm from the surface of the meal to the lens of the camera. The lens formed a 45° angle to the “x”-axis to resemble the view of a plate on a table. The focal length was 32.0 mm, and the shutter speed and aperture were automatically adjusted for each meal. Finally, the lighting conditions were controlled by ensuring total darkness in the room, in which we placed two daylight lamps (E27/55 W) to create optimal lighting conditions.

The pictures were divided into two groups: Group 1 consisted of 10 pictures of complete meals (i.e., vegetable component, staple food, and meat component) varying only in the amount of meat and vegetables. Group 1 was subdivided into two categories: (a) a total of 5 pictures of regular amounts of meat and vegetables and (b) 5 pictures of recommended amounts of meat and vegetables. Group 2 consisted of 50 individual pictures of different portion sizes (extra-small, small, medium, large, and extra-large) of each meat and vegetable ingredient of the meals from Group 1. For instance, regarding the meal of green beans, boiled potatoes, and beef, we took 5 pictures of the portion sizes (extra-small, small, medium, large, and extra-large) of green beans, and 5 pictures of the portion sizes (extra-small, small, medium, large, and extra-large) of beef, resulting in a total of 10 pictures per meal. 

### 2.5. Procedure

Data were collected using EyeQuestion software (Logic8 BV). A survey link was emailed to participants at about local dinner time (between 6:00 p.m. and 7:30 p.m.), which is about five hours after lunch in a regular Dutch schedule. When participants received the email, they followed the instructions on the questionnaire, completed the questions at home, and submitted their responses. The total time to complete the questionnaires was about 20 min. 

### 2.6. Measures

#### 2.6.1. Food Choice between Meals with Regular and Recommended Portions of Meat and Vegetables

Participants were first presented with two pictures (one next to the other) of each meal at a time. One picture corresponded to a meal with regular portions of meat and vegetables, and the other picture corresponded to the same meal but now with recommended portions of meat and vegetables. For each of the five meals, participants were instructed to choose one option of the two portion sizes while being asked the following: “*Imagine it is almost dinner time. You have two options to choose from. Please click on the option that you would like to have for your tonight’s dinner*” (Figure 1). The order of presentation of the portion sizes between meals was balanced among participants. 

#### 2.6.2. Expected Liking, Expected Satiety, and Expected Food-Evoked Emotions 

Right after finishing the food choice task, participants were presented with 10 individual pictures of the regular and recommended portion sizes of each meal one by one in random order. For each individual picture, participants used a 100 mm visual analogue scale (VAS) to evaluate expected liking, expected satiety, and six expected food-evoked emotions (*happy*, *bored*, *disappointed*, *relaxed*, *energetic*, and *satisfied*). For expected liking, the scales were anchored from “*Not liked at all*” to “*Very liked*”; for expected satiety, from “*Not satiated at all*” to “*Very satiated*”; and for expected emotions, from “*Not [emotion] at all*” to “*Very [emotion]*”, e.g., from “*Not happy at all*” to “*Very happy*”.

#### 2.6.3. Perceived Amounts of Meat and Vegetables

Using the 10 individual pictures, participants also evaluated the perceived amounts of meat and vegetables of each portion size (regular and recommended) of the five different meals on a 5-point Likert scale. The scale contained five categories: “*Far too little*”, “*Too little*”, “*Right amount*”, “*Too much*”, and “*Far too much*”. 

#### 2.6.4. Estimations of the Perceived “Normal Portion Size” of Meat and Vegetables

The estimated *“normal”* portion sizes of meat and vegetable ingredients of the five meals were assessed on a 5-point Likert scale. Here, we used the Group 2 pictures (the 50 pictures of meat and vegetable ingredients described in Section 2.4 “Food pictures”) and a sliding scale consisting of five pictures with increasing amounts of food representing the five portion sizes reported in Table 1 for each meat and vegetable ingredient (Figure 2). Participants were asked “*Please move the bar from left to right and release it when you see the amount of food that is a normal amount of food for you to eat at dinner as part of a meal*”. Post hoc, we labelled these five portions as “*extra-small*”, “*small*”, “*medium*”, “*large*”, and “*extra-large*”.

#### 2.6.5. Attitudes and Frequency of Meat and Vegetable Consumption 

At the end, we asked for (self-reported) eating patterns, such as *meat eater, flexitarian, pescatarian, vegetarian*, or *vegan*, and (self-reported) frequency of consumption of vegetables and meat shown in the five dinner meals (”*How often have you consumed any of the following products during the past month?*”), on a 10-point scale from “*Not at all this month*” to “*Everyday*”. The subscale General Health Interest (8 items) from the Health and Taste Attitude Scales was used to measure interest in eating healthily [36]. Furthermore, we assessed the participants’ (self-reported) interest in engaging in reducing meat consumption and increasing vegetable consumption (i.e., which stage of behavioral change they were in), based on the five stages of the Trans Theoretical Model of Behavioural Change [37]. Participants were asked to describe which statement best represented their current interest in reducing meat and increasing vegetable consumption among the following options: stage (1) “*I am not at all interested in reducing meat / increasing vegetable consumption in my diet and I have no intention in doing so in the next 6 months*”; stage (2) “*I am interested in reducing meat / increasing vegetable consumption in my diet and I have the intention of doing that within the next 6 months*”; stage (3) “*I am interested in reducing meat / increasing vegetable consumption in my diet and I have the intention in doing that in the next month*”; stage (4) “*I am interested in reducing meat / increasing vegetable consumption in my diet and I have started reducing meat / increasing vegetable consumption during the last 6 months*”; stage (5) “*I am interested in reducing meat / increasing vegetable consumption in my diet and I have already reduced meat / increased vegetable consumption in my diet for longer than 6 months*”. Participants were then regrouped into three “Interest in Reducing Meat” and “Interest in Increasing Vegetables” groups, i.e., “*not aware, no action*” group (stage 1), “*aware, no action*” group (stages 2 and 3), and “*aware and action*” group (stages 4 and 5).

### 2.7. Data Analysis

Statistical analyses were performed with SPSS (version 25; IBM Corp, New York, NY, USA) and RStudio (R version 4.0.2; Rstudio Team, 2020, Boston, MA, USA). Non-parametric tests were used for all data analyses as visual inspection, and Kolmogorov–Smirnov normality tests indicated that the data were not normally distributed. We considered differences to be significant at *p* < 0.05.

Regarding food choice, a logistic regression with mixed model analysis was performed to assess the probability of participants choosing the recommended over the regular portion of the meals. Moreover, a “McNemar” analysis was conducted to evaluate differences in food choices between meals with regular and recommended portion sizes of meat and vegetables.

For expected liking, expected satiety, and food-evoked emotions, we calculated the median and interquartile range (IQR) of the intensities of expected liking, expected satiety, and the six food-evoked emotions for regular and recommended portion sizes for each meal. A Wilcoxon signed related samples rank test was performed to assess significant differences in expected liking, expected satiety, and the six emotions regarding the regular and the recommended portion sizes in each meal. 

We used chi-squared tests [38,39,40] to evaluate the associations between the proportions of preferred portion size, i.e., regular or recommended, and the self-defined eating patterns, i.e., meat eater, flexitarian, or pescatarian. We also tested the associations between the proportions of preferred portion size and the frequency of meat intake. In addition, we used a chi-squared test to assess the associations between the proportion of preferred portion size and the attitudes to reduce meat intake. Finally, for the perceived amount of meat and vegetables, and gender differences, chi-squared tests were used, and 100% stacked bar plots were used to visualize the perceived amounts of meat and vegetables (from “far too little” to “far too much”) in the regular and recommended portion sizes of each meal.

## 3. Results

### 3.1. Food Choice between Meals with Regular and Recommended Portions of Meat and Vegetables

There were significant differences in food choices between meals with regular and recommended portions of meat and vegetables (*p* < 0.001). Regarding pizza, almost 80% of participants selected the recommended portion size. Regarding fish and carrots, 30% of participants chose the recommended portion. Regarding meals with beef and beans, chicken and peas, and minced meat (Bolognese sauce) and broccoli, 45%, 40%, and 55% of participants chose the recommended portions, respectively. Finally, we observed that women (54%) chose meals with recommended portions more often than men (45%) (*p* < 0.05).

### 3.2. Expected Liking, Satiety, and Food-Evoked Emotions for Regular and Recommended Portions of Meat and Vegetables

Table 2 and Table 3 show the medians and interquartile ranges of the intensities of expected liking, expected satiety, and expected food-evoked emotions for the regular and recommended portion sizes of each meal. The differences between the median intensities were generally small; in all cases, they were below 10 points. Overall, meals with regular and recommended portions were expected to be moderately liked; both scored between 56 and 68 on the 100 mm VAS. For expected satiety, the scores of the recommended and regular portions were similar, in a range between 60 and 70 on the 100 mm VAS. For expected food-evoked emotions, meals with regular and recommended portions scored higher on the positive emotions of *happy*, *relaxed*, *energetic*, and *satisfied* than on the negative emotions of *bored* and *disappointed*. The scores of the expected positive emotions of *happy*, *relaxed*, and *satisfied* ranged between 55 and 66, whereas the scores of the negative emotions of *bored* and *disappointed* were lower, in a range between 29 and 44, on their respective scales. In terms of the positive emotion of *energetic*, both regular and recommended portions of all meals scored above 60, except for pizza. Regarding pizza, the regular and recommended portions scored similarly on *energetic* (51 and 52, respectively, on the 100 mm VAS). Finally, there were no gender differences in food-evoked emotions towards the meals (data not shown).

### 3.3. Perceived Amount of Meat: Too Little or Too Much?

Figure 3 depicts how participants evaluated the perceived amount of meat in the regular and recommended portions in each meal. For each meal, the recommended portion compared with the regular portion shifted in the direction of “too little”. In addition, we found significant effects of portion size (*p* = 0.03) and type of meal (*p* < 0.001) on how participants perceived the amount of meat. Overall, a significant difference was found in the proportions of the perceived amount of meat between the regular and the recommended portions (*X*^2^ (16, N = 1290) = 1527.4, *p* = 0.000). It can also be observed that participants perceived the recommended portion to be (far) too little compared with the regular portions (Figure 3).

In Figure 3, it can be observed that the perceived amount of meat varied not only between regular and recommended portion sizes but also among the types of meal. For instance, the recommended portion of minced meat was more often perceived to be “too little” than the recommended portion of fish. Moreover, the recommended portion of salami was more often perceived to be “too much” than the recommended portion of minced meat. 

There was a clear distinction between males and females in the perception of the amount of meat (Figure 4). A significant difference was found in the proportions of how men and women perceived the amount of meat in the regular portion sizes (*X*^2^ (4, N = 1290) = 62.9, *p* < 0.001). It was observed that more men than women perceived the amount of meat in the regular portions to be “The right amount”. In addition, regarding the recommended portion, a significant difference was found in the proportions of how men and women perceived the amount of meat in the recommended portion sizes (*X*^2^ (4, N = 1290) = 60.7, *p* < 0.001). It was observed that more women than men perceived the amount of meat in the recommended portions to be “The right amount”; on the other hand, more men than women perceived the amount of meat in the recommended portions to be “(Far) too little”.

### 3.4. Perceived Amount of Vegetables: Too Little or Too Much?

Figure 5 shows the perceived amount of vegetables in the regular and recommended portions per meal. Overall, a significant difference was found in the proportions of the perceived amount of vegetables between the regular and the recommended portions (*X*^2^ (16, N = 1290) = 1030.4, *p* < 0.001). It can be observed that the recommended portions were more likely to be perceived to be “(Far) too much” than the regular portions (Figure 5).

Figure 6 shows the amounts of meat and vegetables perceived by males and females. Overall, participants tended to perceive the recommended portion (less meat and more vegetables) to be “(far) too much” compared with the regular portion in each meal. Similar to the amount of meat, the perceived amount of vegetables varied between the regular and the recommended portion sizes and among the types of meal. For instance, carrots were more perceived to be “(far) too much” in the fish meal than tomatoes and peppers in the pizza meal. Opposite to the meat, regarding the recommended portion compared with the regular portion, the evaluation of the amount of vegetables shifted in the direction of “too much”. This was the case for each of the five meals. In terms of the amount of vegetables perceived by men and women, no significant differences were found in the proportions of how men and women perceived the amount of vegetables in meals with regular portion sizes (*X*^2^ (4, N = 1290) = 4.3, *p* = 0.37) nor in meals with recommended portion sizes (*X*^2^ (4, N = 1290) = 3.4, *p* = 0.49).

### 3.5. Estimations of the “Normal Portion Size” of Meat and Vegetables 

Table 4 displays the estimations of the normal portion sizes of the meat and vegetables for males and females. Overall, 64% of males considered the “*regular*” and the “*extra-large*” portions as their “normal” amount of meat, whereas more females (32%) than males (15%) considered the “*recommended*” and the “*extra-small*” portions as their normal amounts of meat. These data are consistent with the results on meat in Figure 4. Regarding vegetables, the gender difference in the estimation of normal portion size was less evident than with meat. The distributions of responses of men and women were quite similar regarding the estimated normal portion size of vegetables, which is consistent with the data in Figure 6. 

### 3.6. Attitudes and Frequency of Meat and Vegetable Consumption 

Table 5 displays the participants’ interest in reducing meat and increasing vegetable consumption. There were significant differences between males and females in terms of the “Interest in Reducing Meat” and “Interest in Increasing Vegetables” groups, with relatively more females in the “aware, and action” group than males, and less females in the “not aware, no action” group than males. We did not observe a significant association between “Interest in Reducing Meat” and “Interest in Increasing Vegetables”, and the perceived optimal amounts of meat and vegetables. Regarding the self-reported frequency of consumption, male and female participants reported similar percentages regarding whether they considered themselves *meat eaters* (57% and 53%, respectively), *flexitarians* (43% and 46%, respectively), *pescatarians* (1% and 1%, respectively), and 0% *vegetarians or vegans*. Overall, females reported to eat meat less frequently than males in the last month (55% versus 72%). In terms of vegetables, the differences were less clear, with, on average, similar frequencies of vegetable intake for women and men (33% versus 31%).

The self-declared eating pattern had an effect on the preferred portion size, i.e., regular or recommended. A significant association was found between the self-reported eating pattern and the preferred portion size (*X*^2^ (2, N = 1290) = 17.36, *p* < 0.001). It was observed that self-declared meat eaters were more likely to prefer the regular portion size, whereas self-declared flexitarians and pescatarians were more likely to prefer the recommended portion size. 

In addition, we tested the relationship between the participants’ attitudes towards reducing meat intake, i.e., “Aware and action”, “Aware and no action”, and “Not aware and no action”, and the preferred portion size, i.e., regular or recommended. No associations were observed between attitudes towards reducing meat consumption and the participants’ preferred portion size for meals with beef (*X*^2^ (2, N = 258) = 4.38, *p* = 0.11), chicken (*X*^2^ (2, N = 258) = 5.16, *p* = 0.08), minced meat (*X*^2^ (2, N = 258) = 2.32, *p* = 0.31), and salami pizza (*X*^2^ (2, N = 258) = 4.29, *p* = 0.12). Only for the meal with fish, a significant association was found between attitudes towards reducing meat consumption and the participants’ preferred portion size (*X*^2^ (2, N = 258) = 7.82, *p* = 0.02). Here, participants who were in the categories of “Aware and action” and “Aware but no action” preferred the regular portion size more than participants in the “Not aware and no action” categories.

Finally, we evaluated the association between the participants’ self-reported frequency of meat intake and their preferred portion size, i.e., regular or recommended. No significant associations were observed between the preferred portion size and the frequency of meat intake (*X*^2^ (3, N = 1290) = 4.92, *p* = 0.18).

## 4. Discussion

In this online study, both the regular and recommended portions of the five meals scored above 55 in expected liking and expected satiety. The positive emotions of *happy, relaxed,* and *satisfied* scored above 55, and negative emotions such as *boredom* and *disappointed* scored below 44 on the 100 mm VAS used. The type of meal influenced the participants’ preference for the regular or recommended portion size. Our outcomes suggest that the effective implementation of portion size strategies for increasing vegetable and limiting meat intake requires a careful, holistic approach focusing on the sensory characteristics of products as well as food-evoked emotions. 

The evaluation of consumer food-evoked emotions adds predictive power to a food choice prediction model compared with a model based on food liking ratings alone [19,20]. It has also been repeatedly shown that liking ratings alone do not accurately predict food choice behavior [41]. To better understand consumer behavior and to implement effective healthy eating strategies, a broader and more holistic perspective on how people experience a food product is, therefore, needed. This perspective goes beyond sensory liking and includes the emotional associations that consumers experience and attach to foods [20]. Such a perspective is relevant from a product development or marketing point of view, but it also helps to better explain consumer perceptions and choice behavior, so as to facilitate new leverage points to change preferred choices into healthier choices. The results of the current study suggest that recommended portion sizes may evoke emotions similar to those evoked by regular portion sizes.

Our results on differences in the optimal amounts of meat and vegetables perceived by men and women are very consistent across the various tasks in the study. Figure 4 shows that men clearly liked larger portions of meat, whereas the difference in gender regarding optimal portions of vegetables was less explicit. These findings are in line with previous studies and consumption surveys [4,31] where men reported a higher meat intake than women. A body of research suggests the importance of the role of gender in meat and vegetable intake. Meat is associated with masculinity, while vegetarianism is strongly associated with femininity [30,42,43,44]. Moreover, the positive attitudes towards increasing vegetable and reduce meat intake in this study shown by women might also explain the higher willingness of women to increase vegetable and reduce meat intake. In contrast, the higher meat consumption frequency and positive attitudes towards meat intake of men participants might also explain the lower willingness to reduce meat consumption [45,46].

Another factor that might currently contribute to women’s greater vegetable and lower meat intake is the association between vegetarianism and highly educated women [36,37,38,39,47,48,49,50]. In addition, employment status [47], personal traits, conscientiousness, and health-related attitudes are related to the association between women and vegetarianism [49]. Finally, the different energy requirements for men and women might have also played a role in the attachment of male participants to larger portions of meat and of women participants to larger portions of vegetables and smaller portions of meat. Typically, women need and consume less calories and protein than men.

Our finding of a significant association between self-defined eating patterns and preferred portion sizes is in agreement with the existing literature [51,52,53]. We observed that meat eaters preferred larger portions containing more meat and less vegetables, while flexitarians and pescatarians opted for portions with less meat and more vegetables. However, as explained by [54], despite the environmental and health benefits of shifting to a diet with less meat and more vegetables, in the Netherlands and Belgium, meat consumption levels continue to be above national dietary guidelines. A flexitarian eating pattern, from a behavioral perspective, may be a more sustainable approach to reducing meat consumption, as it makes gradual changes in eating behavior possible [45,55]. The current results contribute to our understanding of meat reduction. However, much work remains to be conducted in the field of segments of flexitarians, i.e., light and heavy flexitarians, meat abstainers, their motives, and capabilities and challenges to improve the direction of strategies to increase plant-based protein intake.

The results of this online survey suggest that managing portion sizes in a meal is a potential effective tool to increase vegetable intake and reduce meat intake. However, for consumers with positive attitudes towards meat intake, a nudging strategy might be more promising, as concluded by Hartmann and Siegrist [46]. Offering meals that do not need meat in order to be perceived as a complete meal (e.g., pizza or pasta dishes) may be promising in reducing meat intake [46]. This conclusion concurs with our results on pizza, where participants preferred the recommended portion with more vegetables and less meat over the regular portion with less vegetables and more meat. Understanding these differences better might help to design and implement strategies to reduce meat consumption with a psychological and social approach to behavior change. 

Overall, we observed that the results in this study agree with earlier research that suggests that it is possible to offer smaller portion sizes of meat and larger portions of vegetables and that consumers would still remain satisfied [28,29] and pleased. These findings open an opportunity for research in the trending online market, where naturalistic environments and laboratory-controlled conditions could not be achieved. However, further research is needed to validate online results and compare them to actual food choice and intake, actual food perception, and food-evoked emotions.

The results of this online survey are only valid for a Western population in a developed world where people eat more meat and less vegetables than is recommended for a healthy and balanced diet. However, this online survey was conducted in a diverse group of people in the Netherlands, with equal distribution of men and women. Another limitation of this online study is that the device used by participants to complete their questionnaires at home was not standardized; consequently, participants’ responses to the pictures on screen might have been influenced by the different quality of their devices.

## 5. Conclusions

Different meals bring different opportunities to increase vegetable intake, reduce meat intake and meet the nutritional recommendations for healthy eating. Changes in food portion sizes is a potentially effective strategy to reduce meat intake and increase fruit and vegetable consumption. This study showed that recommended portion sizes lead to positive food-evoked emotions, implying that the effective implementation of portion size strategies for increasing vegetable and limiting meat intake requires a careful, holistic approach focusing on the sensory characteristics of food products as well as emotions evoked by the total food experience. 

Our study provides evidence that the type of meal and the self-defined eating pattern are both associated with the preferred portion size. Participants displayed some willingness to increase vegetable intake and reduce meat consumption, although this attitude was dependent on the type of meal. Our findings suggest that a flexitarian eating pattern may be a more sustainable approach to reducing meat consumption, and that further research is needed in order to understand the segments of flexitarians, and their motives and challenges to shift to a more plant-based diet. Finally, it is worthwhile to validate online studies and compare their results with real-life setting conditions.

## Figures and Tables

**Figure 1 foods-12-01259-f001:**
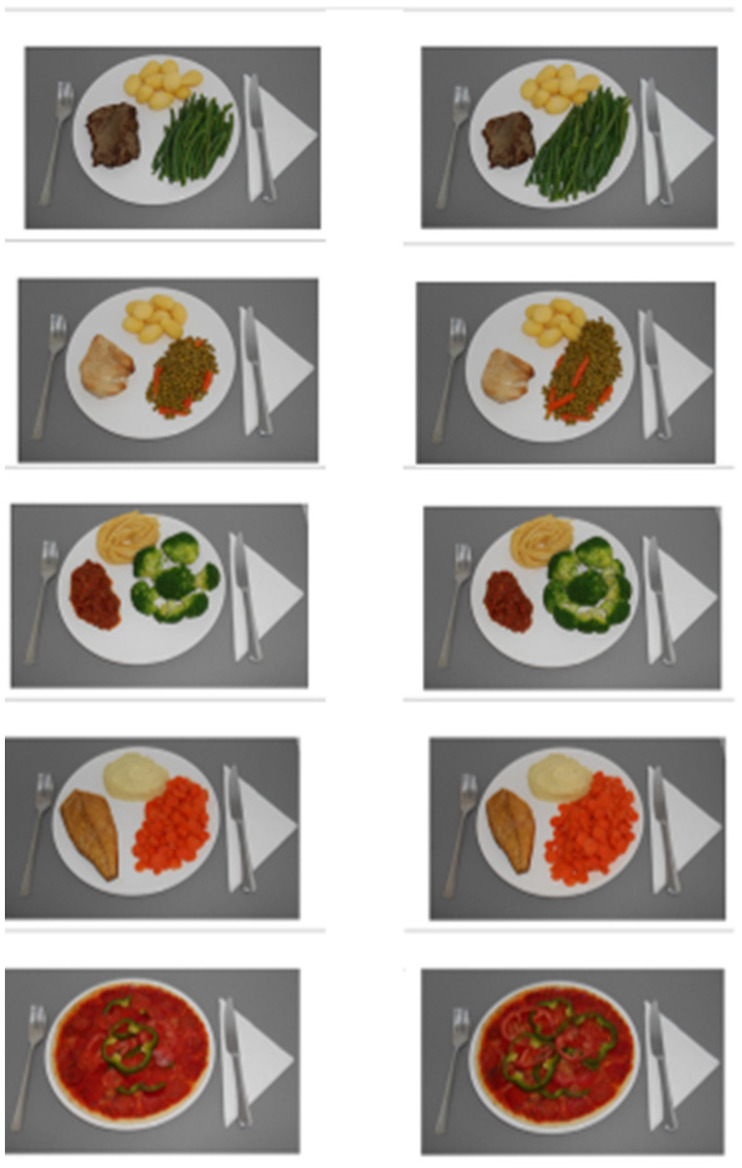
Meals for food choice task. The left pictures reflect meals with regular portions of meat and vegetables, and the right pictures represent meals with recommended portions of meat and vegetables.

**Figure 2 foods-12-01259-f002:**
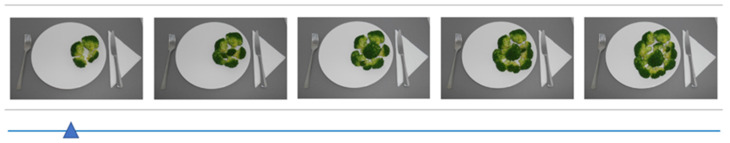
Example of five pictures with increasing portion sizes of broccoli. The sliding triangle was used to estimate the “normal” portion size of each of the five vegetables and each of the five meats.

**Figure 3 foods-12-01259-f003:**
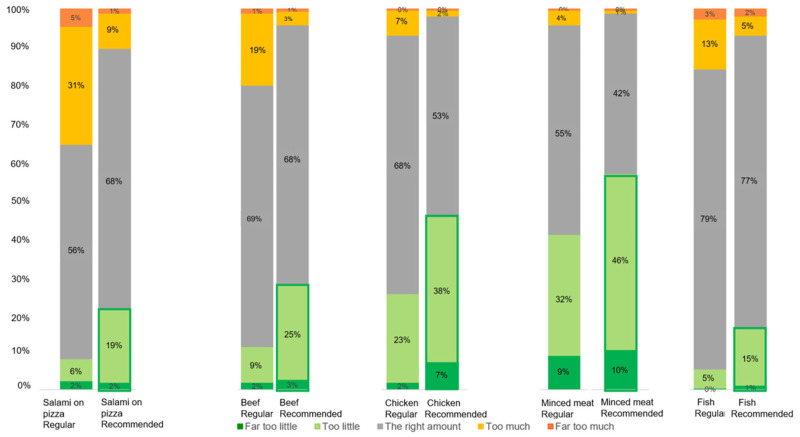
Bar plots of the perceived amount of meat (from “far too little” to “far too much”) in the regular and recommended portions of each of the five meals.

**Figure 4 foods-12-01259-f004:**
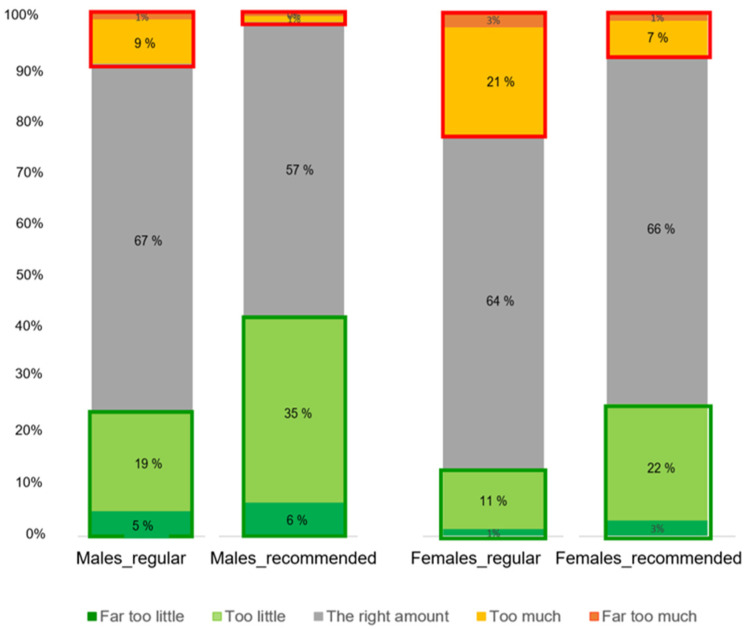
Bar plots of the perceived amount of meat in regular and recommended portion sizes across the five meals by gender.

**Figure 5 foods-12-01259-f005:**
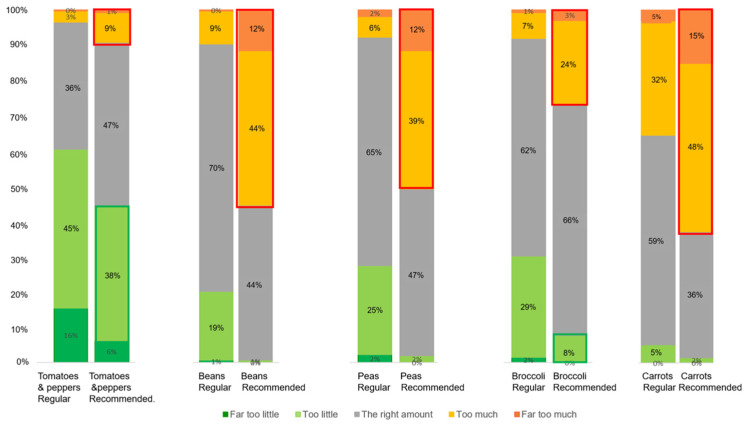
Bar plots of the perceived amount of vegetables (from “far too little” to “far too much”) in the regular and recommended portions of each of the five meals.

**Figure 6 foods-12-01259-f006:**
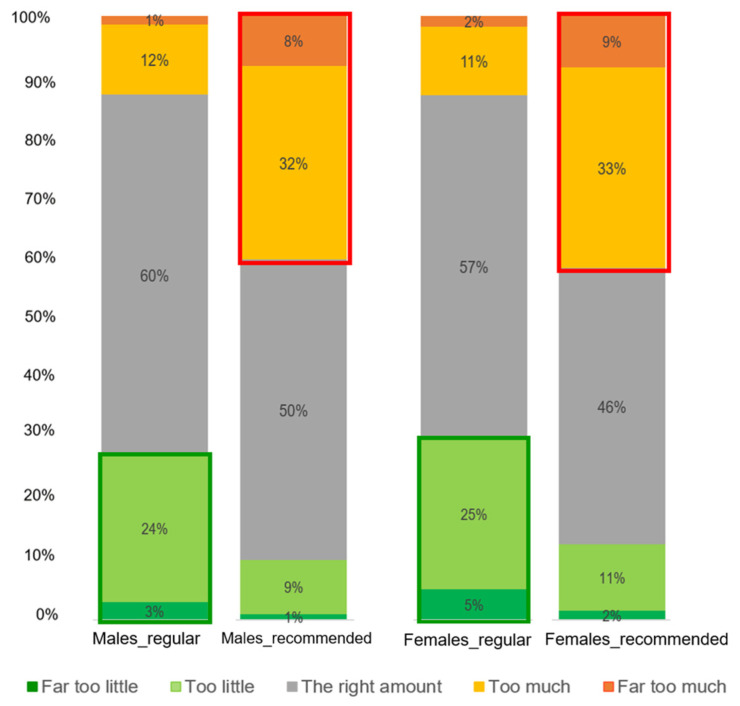
Bar plots of the evaluation of perceived amount of vegetables in regular and recommended portion sizes across the five meals by gender.

**Table 1 foods-12-01259-t001:** Grams of meat and vegetables used per portion size.

	Meat		Reference	Vegetables	Reference
Portion Size	Beef/Chicken/ Minced Meat/Fish	Salami			
Extra-small	75 g	19 g	25% less than the recommended amount of meat per day	97 g	25% less than the average amount of vegetables consumed per day
Small	100 g	25 g	Recommended amount of meat per day [5]	130 g	Average amount of vegetables consumed per day [4]
Medium	112 g	37 g	Between the recommended amount of meat per day and the commercially available amounts of meat	165 g	Between therecommended amount of vegetables per day and theaverage amount of vegetables consumed per day
Large	125 g	50 g	Average portion of meat available in local supermarkets	200 g	Recommended amount of vegetables per day [5]
Extra-large	156 g	63 g	25% more than the average portion of meat available in local supermarkets	250 g	25% more than the recommended amount of vegetables per day

**Table 2 foods-12-01259-t002:** Medians and interquartile ranges of the intensities of expected liking, expected satiety, and expected food-evoked emotions for regular and recommended portion sizes per meal.

PRODUCT	Beef and Beans						Chicken and Peas						Minced Meat (Bolognese) and Broccoli					
SIZE	Regular			Recommended			Regular			Recommended			Regular			Recommended		
	*Mdn*	Q1	Q3	*Mdn*	Q1	Q3	*Mdn*	Q1	Q3	*Mdn*	Q1	Q3	*Mdn*	Q1	Q3	*Mdn*	Q1	Q3
Expected liking	**67**	54	78	65 ***	49	77	56	38	72	56	37	72	63	44	75	62	47	74
Expected satiety	68	58	78	**71 ****	58	83	61	44	74	**64 *****	49	77	60	45	72	62	48	75
Happy	64	51	76	64	49	76	57	39	69	55	37	67	57	44	72	59	44	72
Bored	32	18	50	35	16	53	43	21	59	44	22	60	40	20	56	37	19	57
Disappointed	29	16	47	29	12	47	37	20	55	36	20	55	35	20	57	33	18	54
Relaxed	61	50	72	61	50	74	56	47	68	56	47	70	56	46	70	57	46	71
Energetic	65	52	74	**66 ***	53	77	61	48	72	**60 ***	50	73	61	49	72	**63 ***	51	74
Satisfied	65	51	77	65	52	76	60	44	71	57	43	71	60	51	73	**62 ***	47	74

Significant differences between medians of regular and recommended portion sizes of each meal, using non-parametric related samples of the Wilcoxon rank test. The number in bold indicates the highest number of the significant pairs. * 0.01 < *p* < 0.05, ** 0.001 < *p* < 0.01, and *** *p* < 0.001.

**Table 3 foods-12-01259-t003:** Medians and interquartile ranges of the intensities of the expected liking, expected satiety, and expected food-evoked emotions for regular and recommended portion sizes per meal.

PRODUCT	Fish and Carrots						Pizza Salami with Tomatoes and Peppers					
SIZE	Regular			Recommended			Regular			Recommended		
	*Mdn*	Q1	Q3	*Mdn*	Q1	Q3	*Mdn*	Q1	Q3	*Mdn*	Q1	Q3
Expected liking	62	44	75	61	44	74	65	52	80	**68 *****	57	81
Expected satiety	66	51	78	**68 ***	55	79	68	53	82	70	53	83
Happy	59	45	74	57	44	73	58	42	75	**62 ***	48	76
Bored	34	20	54	**43 ***	21	58	32	17	50	30	15	52
Disappointed	33	16	52	34	17	53	36	19	60	35	17	55
Relaxed	56	48	72	59	47	73	58	48	73	59	48	74
Energetic	63	50	76	61	49	75	51	35	66	52	38	71
Satisfied	**62**	49	76	60 **	45	74	56	41	73	59	44	72

Significant differences between medians of regular and recommended portion sizes of each meal, using non-parametric related samples of the Wilcoxon rank test. The number in bold indicates the highest number of the significant pairs. * 0.01 < *p* < 0.05, ** 0.001 < *p* < 0.01, and *** *p* < 0.001.

**Table 4 foods-12-01259-t004:** Distributions of proportions (%) of the estimated “normal” portion sizes of meat and vegetable ingredients for males and females.

	Meat Ingredients		Vegetable Ingredients	
	Males	Females	Males	Females
Extra-small	2	11	7	11
Recommended	13	21	29	26
Medium	21	25	24	24
Regular	36	29	23	21
Extra-large	28	14	17	19

**Table 5 foods-12-01259-t005:** “Interest in Reducing Meat” groups and “Interest in Increasing Vegetables” groups by gender *.

	Interest in Reducing Meat		Interest in Increasing Vegetables	
	Males	Females	Males	Females
	%	%	%	%
Not aware, no action	29.9	25.2	18.9	15.3
Aware, no action	18.1	13.7	22	17.6
Aware and action	52	61.1	59	67.1

***** The three “Interest in Reducing Meat” groups and “Interest in Increasing Vegetables” groups were based on the five stages of the Trans Theoretical Model of Behavioural Change [37], i.e., *”not aware, no action”* group (stage 1), *”aware, no action”* group (stages 2 and 3), and *”aware and action”* group (stages 4 and 5).

## Data Availability

Research data is stored on the Wageningen University & Research server. Data are available upon request.

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
