# Peer review of "Food-Evoked Emotions and Optimal Portion Sizes of Meat and Vegetables for Men and Women across Five Familiar Dutch Meals: An Online Study"

_foods, 2023, doi:10.3390/foods12061259_

Round 1
Reviewer 1 Report
In my opinion, the paper is interesting, even if not original at all. The analysis is linked at a specific country with particular features compared to those of many other countries. Yet it can be a good reference point for similar studies to be conducted in other countries. A more adequate introduction and literature review could justify the research. Introduction presents properly the aim of the study, yet the research questions to be addressed are not clearly exposed and, above all, justified by the literature. As a matter of fact, the author/s must include accurate and recent references to support the hypotheses and the study. So, strongly I suggest to consider a more recent and innovative papers on the topic and important in the international context. Research design and methodology could be appropriate, yet different analyses have been conducted which enrich the empirical analysis (so again, the authors must consider further literature, I strongly suggest Doi 10.1002/csr.1873, and it must be justified in the text): I recommend the author/s to better specify the goodness of the specific quantitative method to support the conceptual model. And moreover, why is the used methodology better than other important ones? And besides, are the author/s sure that the sample is representative of the population? Especially interesting is the analyses conducted, but I can say also the results could be more appropriate and clear; moreover, discussion section is relevant and conclusions must resume properly the topic address and the implications for several players. So, really what does the paper add to previous researches? The quality of communication is good and clear enough.
Author Response
We would like to thank the reviewers for the constructive comments and suggestions to improve the paper. Please find below our reply. The adjustments improved the paper. In the revised paper we use the font colour red to indicate the changes and additions made.
Reply comments Reviewer #1
Comments:
“In my opinion, the paper is interesting, even if not original at all. The analysis is linked at a specific country with particular features compared to those of many other countries. Yet it can be a good reference point for similar studies to be conducted in other countries. A more adequate introduction and literature review could justify the research. Introduction presents properly the aim of the study, yet the research questions to be addressed are not clearly exposed and, above all, justified by the literature. As a matter of fact, the author/s must include accurate and recent references to support the hypotheses and the study. So, strongly I suggest to consider a more recent and innovative papers on the topic and important in the international context. Research design and methodology could be appropriate, yet different analyses have been conducted which enrich the empirical analysis (so again, the authors must consider further literature, I strongly suggest Doi 10.1002/csr.1873, and it must be justified in the text): I recommend the author/s to better specify the goodness of the specific quantitative method to support the conceptual model. And moreover, why is the used methodology better than other important ones? And besides, are the author/s sure that the sample is representative of the population? Especially interesting is the analyses conducted, but I can say also the results could be more appropriate and clear; moreover, discussion section is relevant and conclusions must resume properly the topic address and the implications for several players. So, really what does the paper add to previous researches? The quality of communication is good and clear enough.”
Reply
We thank the reviewer for the constructive comments and suggestions to improve the paper. Indeed, we consider this paper as a reference point for future studies in other countries.
“Introduction presents properly the aim of the study, yet the research questions to be addressed are not clearly exposed and, above all, justified by the literature. As a matter of fact, the author/s must include accurate and recent references to support the hypotheses and the study. So, strongly I suggest to consider a more recent and innovative papers on the topic and important in the international context”
We appreciate your remarks on the research questions and literature. We included some recent papers to support the hypotheses of the study, and included three research questions in lines 121 – 134 of the revised manuscript.
“Experiments have shown that modifying the portion sizes of meat products can change the food behaviors of consumers without affecting their satisfaction [29, 33]. Also, a recent study found that in the USA consumers are interested in eating less red meat. The study explains that consumers prefer to eat smaller portion sizes of red meat instead of eating meat less often [34]. In terms of gender differences in meat intake. Studies have found that men prefer larger portions. Hence, this online study aims to respond to the following research questions:
- a) Is there any preference for the regular or recommended portion size of the offered meals
- b) Is there any difference between regular and recommended portions in the intensity of positive and negative food-evoked emotions.
- c) Is there any gender difference in the perception of the amount of meat and vegetables.”
Research design and methodology could be appropriate, yet different analyses have been conducted which enrich the empirical analysis (so again, the authors must consider further literature, I strongly suggest Doi 10.1002/csr.1873, and it must be justified in the text):
Thank you for your suggestions about the appropriate literature to support our methodology and analysis. We added a number of Chi-squared tests to analyse the data, and cited Doi 10.1002/csr.1873 in this context. We included the following paragraph in the manuscript.
Lines 329 - 335
“We used Chi-Squared Tests [38-40] to evaluate the associations between the proportions of preferred portion size, i.e. regular or recommended, and the self-defined eating patterns, i.e. meat eater, flexitarian, pescatarian. We also tested the association between the proportions of preferred portion size and the frequency of meat intake. In addition, we used a Chi-Squared test to assess the association between the proportion of preferred portion size and the attitudes to reduce meat intake. Finally, for the perceived amount of meat and vegetables and gender differences Chi-Squared Tests were used and […].”
I recommend the author/s to better specify the goodness of the specific quantitative method to support the conceptual model. And moreover, why is the used methodology better than other important ones?. And besides, are the author/s sure that the sample is representative of the population?
The conceptual method allowed us to collect data in the direction of more or less preferred quantities of meat and vegetable. We now use the X-squared statistic as a quantitative evaluation of differences between consumer segments. In our case, we controlled for variables such as gender, age, region, eating habits; and measured food choice, intensity of food evoked emotions, perceived amount of food, and frequency of food intake. The market research agency was requested to select 270 participants with equal number of men, women, and the three age groups 18-35, 36-50, and 51–65 y. Subjects came from the west, middle and east of the Netherlands.
We adapted these details of participants in lines 137 - 143
“Subject were recruited by a market research agency Essensor BV in the Netherlands, with the aim to collect data from 270 participants, 50% male and 50% female; aged: 33% between 18 - 35 years, 33% between 36 - 50 years, 33% between 51 - 65 years. Subjects reported to be a meat eater or flexitarian and were from the areas of Utrecht, Wageningen and Vlaardingen (The Netherlands), 33% from each region.”
Especially interesting is the analyses conducted, but I can say also the results could be more appropriate and clear; moreover, discussion section is relevant and conclusions must resume properly the topic address and the implications for several players.
We appreciate that you find the analyses conducted as interesting. In the revised manuscript we include outcomes from the associations between proportions of the preferred portion size, i.e. regular or recommended and the self-defined eating patterns, i.e. meat eater, flexitarian, pescatarian. We included results from Chi-squared tests to underpin the relationships in the various figures and table of the revised manuscript. These include sections in the results sections with lines 421-423, 439-447, 460-464, 473-477, 535-555.
Regarding the discussion section we replaced the word “representative” by “diverse”. Line 639.
We included the following discussion paragraph in lines 603 – 615 to specify the implications of the results for the several players/consumer segments.
“Our findings of a significant association between self-defined eating pattern and preferred portion size are in agreement with existing literature [51-53]. We observed that meat eaters preferred larger portions containing more meat and less vegetables, while flexitarians and pescatarians opted for portions with less meat and more vegetables. However, as explained by [54] despite environmental and health benefits of shifting to a diet with less meat and more vegetables, in the Netherlands and Belgium meat consumption levels continue to be above national dietary guidelines. A flexitarian eating pattern, from a behavioural perspective, may be a more sustainable approach to reducing meat consumption, as it allows for gradual changes in eating behaviour [45, 55]. The current results contribute to our understanding of meat reduction. However much work remains to be done in the field of segments of flexitarians i.e. light and heavy flexitarians, meat abstainers, their motives, capabilities and challenges to improve the direction of strategies to increase plant-based protein intake.”
What does the paper add to previous researches? The quality of communication is good and clear enough.
Regarding the added value of this study compared to previous studies, it is in the measurement of the (expected) food-evoked emotions in relation to optimal portion sizes of meat/vegetables, see lines 95-98 of the introduction and the second part of the revised discussion.

Reviewer 2 Report
The paper “Food-evoked emotions and optimal portion sizes for meat and vegetables for men and women across five familiar Dutch meals: An online study” contributes to the growth of literature for specialists in sensory analysis and nutritionists researching the reduction of meat consumption and the problem of satiety of meals.
The following items should be revised:
Introduction
Line 33-34
“In terms of meat, global meat consumption (excluding white meat) has sharply increased between 1996 and 2016 [6, 7] reaching 360 M tonnes globally in 2018 [7]”.
In recent years, there has been an increase in interest in plant-based diets. For example, from 2019 to 2020, as many as 5 million consumers in the United States shifted to avoid meat altogether, becoming either vegetarians or vegans. - Other countries are the same. - I suggest writing about it.
I suggest writing that the problem of high meat consumption for selected groups of consumers is still valid. Is there any recent information on this area (e.g. 2020-2021)?
Methods
What was the selection of the group? Whether included the exclusion criteria, e.g. pregnancy of women, diseases or lack of them?
What was the interval between the last meal and the examination?
Results and statistical analysis
The results are interesting but no answer to the following :
Is a relationship between the results obtained and the declared diet (meat eater and flexitarian)? e.g. with the chi-squared test
Is there a relationship between the results obtained and the frequency of meat consumption (this may be crucial)? e.g. with the chi-squared test
Results
Figure 3- Figure 6
There are no results for the significance of differences. I suggest you show graphs with the significance of the differences.
Discussion
“The results of this online survey are only valid for a Western population in a developed world where people eat more meat and less vegetables than is recommended for a healthy and balanced diet”
? applies to the Netherlands
? “flexitarian (respectively male 43% and female 46%) - Do they also eat a lot of meat?
“..this online survey was conducted in a representative group of people in the Netherlands,” the description of the group in the methodology does not confirm this, e.g. the maximum age is 52 years and, the minimum age is 24 years. - I suggest removing this sentence.
Conclusion
Similarly to the results - Is a relationship between the results obtained and the declared diet (meat eater and flexitarian)?
Is there a relationship between the results obtained and the frequency of meat consumption (this may be crucial)? - in one sentence.
Author Response
We would like to thank the reviewers for the constructive comments and suggestions to improve the paper. Please find below our reply. The adjustments improved the paper. In the revised paper we use the font colour red to indicate the changes and additions made.
Reply comments Reviewer #2
We would like to thank the reviewer for the constructive comments and suggestions to improve the paper. We were happy to hear that the reviewer found that the study contributes to the growth of literature for specialists in sensory analysis and nutrition. Please find below our reply to the specific comments. We use the font colour red to indicate the changes and additions made in the revised paper.
Introduction
Line 33-34
“In terms of meat, global meat consumption (excluding white meat) has sharply increased between 1996 and 2016 [6, 7] reaching 360 M tonnes globally in 2018 [7]”.
In recent years, there has been an increase in interest in plant-based diets. For example, from 2019 to 2020, as many as 5 million consumers in the United States shifted to avoid meat altogether, becoming either vegetarians or vegans. - Other countries are the same. - I suggest writing about it.
I suggest writing that the problem of high meat consumption for selected groups of consumers is still valid. Is there any recent information on this area (e.g. 2020-2021)?
We appreciate these suggestions and we expanded the introduction to explain this more fully, lines 34-46
“ In terms of meat consumption, literature reports a global increase of 58% of meat intake over 20 years to 2018 in the United States, China and Australia [6]. Moreover, it is expected that the global meat consumption increases by 14% over the next decade compared to a base period average of 2018-2020 [7]. On the other hand, the global concern around consumption of animal products and the awareness of meat intake reduction have led to an increased interest in plant-based diets or change in eating patterns [8-10]. For example, in the United States about 4% of the adults reported being vegan or vegetarians between 2019 and 2020, whereas about 46% reported to sometimes or always eat vegan or vegetarian when eating out [11]. Also, in the Netherlands the red meat consumption decreased from an average of 82 g per day to 79 g per day when comparing the results from the consumption survey of 2012-2016 [4] and 2019-2020 [12]. Interestingly, consumption of poultry increased from average 16 g per day to 18.4 g per day in the same period [4, 12].”
Methods
What was the selection of the group? Whether included the exclusion criteria, e.g. pregnancy of women, diseases or lack of them?
In this study we did not use pregnancy or breastfeeding as exclusion criteria; candidates who reported chewing problems and/or were self-reported as vegan or vegetarians were excluded during the screening
Lines 139 - 143
“Subjects were recruited by a market research agency Essensor BV in the Netherlands, with the aim to collect data from 270 participants, 50% male and 50% female; aged: 33% between 18 - 35 years, 33% between 36 - 50 years, 33% between 51 - 65 years. Subjects reported to be a meat eater or flexitarian and were from the areas of Utrecht, Wageningen and Vlaardingen (The Netherlands), 33% from each region.”
What was the interval between the last meal and the examination?
Participants were asked to fill in this questionnaire at dinner time between 18:00-19:30 which is about five hours after lunch in a regular Dutch schedule.
Lines 216 - 218
“Data was collected using the EyeQuestion software (Logic8 BV). A survey link was emailed to participants at about local dinner time (between 6:00 pm and 7:30 pm) which is about five hours after lunch in a regular Dutch schedule.”
Results and statistical analysis
The results are interesting but no answer to the following :
Is a relationship between the results obtained and the declared diet (meat eater and flexitarian)? e.g. with the chi-squared test
We expanded the results with the following analysis:
Lines 535 - 539
“The self-declared eating pattern had an effect on the preferred portion size, i.e. regular or recommended. A significant association was found between the self-reported eating pattern and the preferred portion size, X2 (2, N = 1290) = 17.36, p < 0.001. It was observed that self-declared meat eaters were more likely to prefer the regular portion size, meanwhile the self-declared flexitarians and pescatarians were more likely to prefer the recommended portion size. ”
Lines 541 - 555
“Also, we tested the relationship between the participants’ attitudes towards reducing meat intake, i.e. “Aware and action”, “Aware and no action” and “Not aware and no action” and the preferred portion size, i.e. regular or recommended. No association was observed between attitudes towards reducing meat consumption and participant’s preferred portion size for meals with beef X2 (2, N = 258) = 4.38, p = 0.11, chicken X2 (2, N = 258) = 5.16, p = 0.08, minced meat X2 (2, N = 258) = 2.32, p = 0.31 and salami pizza X2 (2, N = 258) = 4.29, p = 0.12. Only for the meal with fish, a significant association was found between attitudes towards reducing meat consumption and participant’s preferred portion size X2 (2, N = 258) = 7.82, p = 0.02. Here, participants who are in the categories “Aware and action” and “Aware but no action” preferred the regular portion size more than participants from the “Not aware and no action categories.”
Is there a relationship between the results obtained and the frequency of meat consumption (this may be crucial)? e.g. with the chi-squared test
This is indeed an interesting question. For results see text in revised manuscript below.
Lines 552 - 555
“Finally, we evaluated the association between participant’s self-reported frequency of meat intake and their preferred portion size, i.e. regular or recommended. No significant association was observed between the preferred portion size and the frequency of meat intake X2 (3, N = 1290) = 4.92, p = 0.18.”
Results
Figure 3- Figure 6
There are no results for the significance of differences. I suggest you show graphs with the significance of the differences.
We appreciate your suggestion. We added the statistical information about the figures in their accompanying texts in the results section. This makes the results section more clear and meaningful. See text sections below.
Lines 421 - 423
“Overall, a significant difference was found in the proportions of the perceived amount of meat between the regular and the recommended portions X2 (16, N = 1290) = 1527.4, p = 0.000.”
Lines 439 - 447
A significant difference was found in the proportions about how men and women perceived the amount of meat in the regular portion sizes X2 (4, N = 1290) = 62.9, p < 0.001. It was observed that more men than women perceived the amount of meat in the regular portions as “The right amount”. Also, for the recommended portion, a significant difference was found in the proportions on how men and women perceived the amount on meat in the recommended portion sizes X2 (4, N = 1290) = 60.7, p < 0.001. It was observed that more women than men perceived the amount of meat in the recommended portions as “The right amount”, meanwhile, more men than women perceived the amount of meat in the recommended portions as “(Far) too little”..
We also expanded the explanation of the results from the perceived amount of vegetables in the regular and recommended portions of each of the five meals in lines 460 - 464 and the perceived amount of vegetables by gender in lines 470 – 474
Lines 460 - 464
“Overall, a significant difference was found in the proportions of the perceived amount of vegetables between the regular and the recommended portions X2 (16, N = 1290) = 1030.4, p < 0.001. It can be observed that the recommended portions were more likely to be perceived as “(Far) too much” as compared to the regular portions (Figure 5).
Lines 473 - 477
“In terms of the perceived amount of vegetables by men and women, no significant difference was found in the proportions of how men and women perceived the amount of vegetables for meals in the regular portion size X2 (4, N = 1290) = 4.3, p = 0.37, nor for meals in the recommended portion size X2 (4, N = 1290) = 3.4, p = 0.49.”
Discussion
“The results of this online survey are only valid for a Western population in a developed world where people eat more meat and less vegetables than is recommended for a healthy and balanced diet”
? applies to the Netherlands
Thank you for this question. Our results apply to the Netherlands where meat consumption levels continue to be far above national dietary guidelines and the EAT-Lancet Commission dietary recommendations (Schuurman, R., 2020, Willett et al., 2019).
? “flexitarian (respectively male 43% and female 46%) - Do they also eat a lot of meat?
We appreciate your question about the amount of meat by the self-declared flexitarian participants. About 58% of the self-declared flexitarian men and 55% of the self-declared flexitarian women reported a meat intake of zero to three days per month. About 42% and 39% of self-declared flexitarian men and women respectively, reported meat intake of one to three days per week. Finally, 2% of the self-declared flexitarian both men and women reported a meat intake four to six days per week, and no flexitarians reported eating meat daily.
This paragraph is not included in the revised manuscript.
“..this online survey was conducted in a representative group of people in the Netherlands,” the description of the group in the methodology does not confirm this, e.g. the maximum age is 52 years and, the minimum age is 24 years. - I suggest removing this sentence.
We thank you for the suggestion. We changed “representative group” into “diverse group” in line 639 and expanded the discussion from lines 603 – 615.
Conclusion
Similarly to the results - Is a relationship between the results obtained and the declared diet (meat eater and flexitarian)?
Is there a relationship between the results obtained and the frequency of meat consumption (this may be crucial)? - in one sentence.
We appreciate your questions to improve the conclusions of this article regarding the relationships between self-declared eating pattern and preferred portion size, as well as the relationship between frequency of meat intake and preferred portion size. We included the following text in lines 656 – 663
Lines 656 - 663
“Our study provides evidence that the type of meal and the self-defined eating pattern are both associated with the preferred portion size. Participants displayed some willingness to increase vegetable intake and reduce meat consumption, although this attitude was dependent on the type of meal. Our findings suggest that a flexitarian eating pattern may be a more sustainable approach to reducing meat consumption, and that further research is needed in order to understand the segments of flexitarians and their motives and challenges to shift to a more plant-based diet. Finally, it is worthwhile to validate online studies and compare their results with real-life setting conditions”.

Round 2
Reviewer 1 Report
Now I am satisfied.
Reviewer 2 Report
I have suggested entries of the "significance" in the figures (as data labels), but I agree with such a notation (in the text).